# Endometriosis in Adolescents with Obstructive Anomalies of the Reproductive Tract

**DOI:** 10.3390/jcm12052007

**Published:** 2023-03-03

**Authors:** Karina Kapczuk, Weronika Zajączkowska, Klaudyna Madziar, Witold Kędzia

**Affiliations:** 1Division of Gynecology, Poznan University of Medical Sciences, Polna 33, 60-535 Poznan, Poland; 2Gynecology and Obstetrics Clinical Hospital of Poznan University of Medical Sciences, Polna 33, 60-535 Poznan, Poland

**Keywords:** endometriosis, genital tract anomalies, Müllerian anomalies, adolescents, OHVIRA syndrome, unicornuate uterus, cervical aplasia

## Abstract

Background: This study aimed to assess the prevalence and course of endometriosis in adolescents with obstructive Müllerian anomalies. Methods: The study group involved 50 adolescents undergoing surgeries (median age 13.5 (range 11.1–18.5)) for rare obstructive malformations of the genital tract: 15 girls had anomalies associated with cryptomenorrhea and 35 were menstruating. The median follow-up period was 2.4 (ranging from 0.1 to 9.5) years. Results: We diagnosed endometriosis in 23 of the 50 subjects (46%), including 10 of the 23 patients (43.5%) with obstructed hemivagina ipsilateral renal anomaly syndrome (OHVIRAS), six of eight patients (75%) with a unicornuate uterus with a non-communicating functional horn, two of three patients (66.7%) with distal vaginal aplasia, and five of five patients (100%) with cervicovaginal aplasia. Persistent dysmenorrhea, following treatment, affected 14 of the 50 adolescents (28%), including 8 of the 17 subjects (47.1%) diagnosed with endometriosis at the time of surgical correction and six adolescents diagnosed with endometriosis during the follow-up. Conclusions: Endometriosis affects about half of young adolescents undergoing surgical treatment of obstructive Müllerian anomalies after menarche. The incidence of endometriosis is highest in girls with cervical aplasia. The risk of developing endometriosis decreases after surgical correction of obstruction but is still significant in patients with uterine anomalies.

## 1. Introduction

Obstructive reproductive tract anomalies and increased retrograde menstruation are well-known risk factors for pelvic endometriosis [1]. These anomalies usually appear in adolescence and in girls, with or without menses. Longitudinal obstructing vaginal septum (obstructed hemivagina), unilateral cervical aplasia, Robert’s uterus, obstructed uterine horn, and accessory cavitated uterine mass (ACUM) [2] are diagnosed in the presence of menstrual cycles. Imperforate hymen, transverse vaginal septum, partial vaginal aplasia, and cervical aplasia [3] are diagnosed in the absence of menses. A study on endometriosis in adolescence confirmed the diagnosis earlier in the subgroup with genital malformations [4]. The study involving 186 patients with Müllerian anomalies (mean age 23.1) revealed that endometriosis was significantly more common in patients with obstructive anomalies as compared with non-obstructive anomalies (57.6% vs. 17.6%, respectively) [5]. The data on the incidence of endometriosis in adolescents with genital malformations leading to an outflow obstruction are limited. Some authors believe that adolescent endometriosis may be distinct from adult endometriosis [6]. Therefore, extrapolating the data from the adult population with delayed diagnosis may be misleading. The issue of the natural course of endometriosis in patients after the repair of obstructive Müllerian anomalies in adolescence is even more unknown, and both scenarios, lesions resolution and persistence, were reported [7,8]. In a study that included 85 adolescents with endometriosis (44 had genital tract malformations), 20% declared no change or a worsening of pain after surgery [9].

The study aimed to assess the prevalence and the course of endometriosis in adolescents diagnosed with and surgically treated for obstructive Müllerian anomalies. Following our institution’s ethics guidelines, approval to conduct this study was not required as the study consisted of a retrospective analysis of the medical records. All patients and their parents were informed that the patients’ medical records might be used for research and scientific publications and signed informed consent forms.

## 2. Materials and Methods

The study group involved 50 adolescent girls that underwent surgical treatment for congenital obstructive anomalies of the genital tract (other than imperforate hymen) in the Division of Gynecology, Gynecology and Obstetrics Clinical Hospital of Poznan University of Medical Sciences (tertiary level hospital), Poland, between January 2012 and June 2021 and were followed-up in Outpatient Clinic of Pediatric and Adolescent Gynecology of this hospital. Fifteen patients manifested anomalies associated with cryptomenorrhea and thirty-five patients displayed obstructive anomalies in the presence of menstrual cycles (in nineteen patients, obstruction occurred on the right side, in sixteen patients, it occurred on the left side).

In each patient, a detailed diagnosis of the genital or urogenital anomaly was made before planning the treatment. The evaluation involved the following: a gynecological examination performed in an adjusted manner, abdominal or transrectal 2D ultrasound (US), and pelvic magnetic resonance imaging (MRI). The urinary tract was investigated by abdominal 2D ultrasound and X-ray or MRI urography in patients who were not previously diagnosed with renal malformations. In the present study, we categorized the anomalies according to the European Society of Human Reproduction and Embryology and the European Society for Gynecological Endoscopy (ESHRE/ESGE) classification system of female genital tract congenital anomalies [10], as well as the American Society for Reproductive Medicine Müllerian anomalies classification (ASRM MAC) 2021 [11]. We analyzed the follow-up data recorded in medical charts up to December 2021 with particular attention to the following: menstrual pain, need for treatment of menstrual pain with non-steroidal anti-inflammatory drugs (NSAIDS), combined hormonal contraception (CHC) or progestogens, diagnosis of endometriosis during follow-up, and family history of endometriosis. We diagnosed pelvic endometriosis based on the results of US and MRI (performed before the surgical treatment of genital anomaly or during follow-up) or positive laparoscopy (performed as a part of the main surgical treatment or during follow-up). We assessed the stage of endometriosis according to the revised ASRM classification of endometriosis [11].

The statistical analysis was performed using STATISTICA10PL. In order to compare the groups, Mann–Whitney non-parametric test was applied. *P* < 0.05 was considered statistically significant. 

## 3. Results

The detailed characteristics of the subjects are presented in Table 1 and Table 2. We found coexisting renal anomalies in 31 patients (62%).

The median age at surgery was 13.5 (range of 11.1–18.5) and the follow-up period ranged from 0.1 to 9.5 (median 2.4) years. The median time from symptoms onset (in patients without menses) or menarche (in patients with menses) to the correction of the malformation was one year (range of 0–7.6, when 0 means less than one month). In two girls (no. 7 and 9) genital anomaly was detected in childhood and these patients were asymptomatic when treated in advanced puberty. We diagnosed endometriosis in 23 subjects (46%): in 12 of 35 patients (34.3%) with obstructive vaginal anomalies and in 11 of 15 patients (73%) with obstructive uterine anomalies. We diagnosed endometriosis (1) in five of five patients (100%) with cervicovaginal aplasia (ESHRE/ESGE: cervical and vaginal aplasia (C4V4); ASRM MAC 2021: cervical agenesis), (2) in two of three patients (66.7%) with distal vaginal aplasia (ESHRE/ESGE: vaginal aplasia (V4); ASRM MAC 2021: distal vaginal aplasia), (3) in 10 of 23 patients (43.5%) with obstructed hemivagina ipsilateral renal anomaly syndrome (OHVIRAS)(ESHRE/ESGE: bicorporeal uterus, double cervix, and longitudinal obstructing vaginal septum (U3C2V2); ASRM MAC 2021: uterus didelphys and obstructed hemivagina), and (4) in six of eight patients (75%) with a unicornuate uterus with a non-communicating functional horn (ESHRE/ESGE: hemiuterus with a rudimentary cavity in not communicating horn (U4a); ASRM MAC 2021: unicornuate uterus with uterine remnant with functional endometrium). One patient with unilateral cervical aplasia (ESHRE/ESGE: bicorporeal uterus, unilateral cervical aplasia, and a normal vagina (U3C3V0); ASRM MAC 2021: uterus didelphys with unilateral cervical atresia) had secondary hypothalamic amenorrhea for three years from menarche up to the surgery and, in contrast to the patients with cervicovaginal aplasia, had no endometriosis. None of the seven patients with the transverse vaginal septum (ESHRE/ESGE: normal uterus, normal cervix, and transverse vaginal septum (U0C0V3); ASRM MAC 2021: transverse vaginal septum) were diagnosed with endometriosis or reported dysmenorrhea during the follow-up.

In the majority of the cases (17 patients, 73.9% of patients with endometriosis, and 34% of the total study group), we diagnosed endometriosis at the time of the surgical treatment of Müllerian anomaly (Table 3). All these patients were treated with dienogest (daily dose 2 mg) for six months after surgery. Post-surgery dysmenorrhea resistance to NSAIDS and requiring medical assistance (long-term treatment with CHC, progestogens, or laparoscopy) affected fourteen adolescents (28%): one patient (1/5, 20%) with cervicovaginal aplasia, one patient (1/3, 33.3%) with distal vaginal aplasia with the coexisting uterine anomaly, seven patients (7/23, 30.4%) with OHVIRAS, and five patients (5/8, 62.5%) with a unicornuate uterus. Persistent pelvic pain affected 8 of 17 patients (47.1%) diagnosed with endometriosis during surgical correction of genital tract anomaly and six adolescents in which we diagnosed endometriosis during the follow-up (Table 4). These six adolescents constituted 18.2% of the subjects without endometriosis at correction and 26.1% of patients with endometriosis. We found moderate to severe endometriosis (stage 3 or 4 rASRM) in seven adolescents (30.4% of subjects affected with endometriosis and 14% of the total study group) (Table 3 and Table 4). Six adolescents (26.1% of patients with endometriosis) had ovarian endometrioma ≥ 4 cm in diameter. None of our subjects were diagnosed with deep infiltrating endometriosis or adenomyosis. Only three patients (6%, no. 1, 16, and 41) had a positive family history of endometriosis.

No significant differences were observed between (1) adolescents diagnosed with endometriosis and without endometriosis, (2) girls diagnosed with endometriosis at surgical correction and during follow-up, and (3) girls with persistent dysmenorrhea and without persistent dysmenorrhea for, (1) the age at surgery, (2) age at the onset of symptoms (in patients without menses) or age at menarche (in patients with menses), and (3) delay from symptoms onset (in patients without menses) or menarche (in patients with menses) (Table 5). Adolescents with persistent dysmenorrhea were older at the last follow-up visit than adolescents without menstrual pain (*p* = 0.004). The duration of follow-up was longer in girls with endometriosis (*p* = 0.027) and in adolescents with persistent dysmenorrhea (*p* = 0.001) (Table 5).

## 4. Discussion

We present an extensive group of young adolescent patients with rare obstructive reproductive tract anomalies, which are analyzed for endometriosis. The true prevalence of pelvic endometriosis in the general population of adolescent and young adult women (under the age of 20) remains unknown [13]. In a study of the Finnish population, published in 2022, the incidence rate of surgically confirmed endometriosis in adolescents (1.5% had associated gynecological malformations) varied from 5.6 to 11.4 per 100,000 persons per years [14]. It should be emphasized that adolescents aged < 17 years comprised only 8.2% of the study group [14]. Our study showed that adolescents with congenital obstructive anomalies of the genital tract, even those treated soon after menarche (median delay was one year), are at high risk of pelvic endometriosis. The overall incidence of pelvic endometriosis in our patients undergoing correction of obstructive reproductive tract anomalies at the median age of 13.5 and followed up to the median age of 16.5 was 46%. We diagnosed endometriosis in three-fourths of our patients during surgical treatment of the obstructive anomaly and in the remaining one-fourth during the follow-up. We also showed that early surgical treatment of obstructive Müllerian anomalies and concomitant endometriosis, if ascertained, resulted in the complete resolution of dysmenorrhea in about 70% of the patients. However, despite proper surgical corrections of the outflow obstruction, in about 18% of the patients without endometriosis at correction, endometriosis displayed during the follow-up. Our findings confirmed that both the incidence and the course of endometriosis are anomaly-specific.

OHVIRAS was the most common anomaly in our study group. A systemic review enrolling 724 patients (from 133 studies) with OHVIRAS reported endometriotic lesions in only 13.6% of patients. However, the authors emphasize underestimating that percentage, because in most patients, a routine laparoscopy was not performed [15]. In a report by Hur et al. [16], severe endometriosis (superficial and deep endometriotic lesions, dense adhesions) was found in a 15-year-old girl with OHVIRAS without symptoms of endometriosis prior to laparoscopy. Laparoscopy or laparotomy was performed in fifteen of our patients with OHVIRAS: endometriosis was found in ten patients. Only two of these girls were diagnosed with ovarian endometrioma preoperatively.

In the study published by Tong et al. [17], the incidence of pelvic endometriosis in patients with OHVIRAS (the authors used the term Herlyn–Werner–Wünderlich syndrome, HWWS) was 19.2% (18/94), which was lower than in our study (43.5% (10/23)). All of these patients (100%) had ovarian endometrial cysts ipsilateral to the obstructed hemivagina, which, in turn, was uncommon for our OHVIRAS patients with endometriosis (20%, 2/10). Nevertheless, four of our patients were diagnosed with advanced endometriosis (stage 3 or 4 rASRM) (40%, 4/10). In the Chinese group, the diagnosis of severe ovarian endometriosis preceded the diagnosis of HWWS, and the patients were older at the diagnosis of HWWS than in our study group [17]. The findings suggest that early treatment of the obstructed longitudinal vaginal septum might significantly reduce the risk of the development of advanced pelvic endometriosis, which is critical for future fertility. Following surgical correction (and six months of post-operative medical therapy in endometriosis-positive cases), 30.4% of our patients with OHVIRAS required further treatment with progestogens or CHC for persistent dysmenorrhea.

Unlike patients with an obstructive vaginal septum and concomitant uterine anomaly, we have not found endometriosis in our adolescent patients with an isolated obstructive transverse vaginal septum. Patients with imperforate hymen have not been considered in our analysis, as these are usually managed during emergencies without extensive clinical evaluation or long-term follow-ups. However, considering that an imperforate hymen represents an even more distant obstruction than transverse vaginal septum, we can assume that both these anomalies do not pose a risk of pelvic endometriosis when treated in early adolescence. Distal vagina aplasia, on the contrary, seems to increase the likelihood of developing endometriosis. However, it should be emphasized that our patient (no. 6) with distal vaginal aplasia and a diagnosis of severe endometriosis during the follow-up period had a concomitant uterine anomaly.

In the present study, we observed the highest prevalence of endometriosis, considering the proportion of patients affected, in girls with cervicovaginal aplasia. Fedele et al. found endometriosis and pelvic adhesions in, respectively, 58% and 83% (7 and 10/12) of adolescents with cervical atresia and vaginal aplasia that underwent laparoscopically assisted uterovestibular anastomosis [18]. Song et al. found pelvic endometriosis in 56% (54/96) of Chinese patients with congenital cervical atresia, with the predominance of ovarian endometrial cysts over peritoneal endometriosis (76% vs. 22% of the patients, respectively) [19]. In the aspect of the preservation of the uterus without re-obstruction, the outcome of cervical atresia treatment was more favorable in our case series (100% of the patients) than in the Chinese study (21.9% of the patients [19]), but the occurrence of endometriosis in our group was higher (100%). Song et al. identified the time, from symptom (pain) onset to surgery, as the only independent predictor of pelvic endometriosis [19]. In our patient (no. 2) who underwent surgery early (6 months after the onset of pain), endometriosis was not found at the time of surgery, but it appeared three years later, despite a patent cervical canal. On the other hand, in a patient (no. 1) who underwent surgery 2.5 years after the onset of symptoms (the most prolonged delay in our study group), minimal endometriosis was diagnosed during surgery, but the long-term follow-up was uneventful. In this patient, menstrual suppression was applied, first with the use of CHC and then aGnRH; the surgery had to be postponed for psychological reasons. It is noteworthy that an estimation of the onset of symptoms suggesting menarche in patients with a complete obstruction of the genital tract can entail significant errors and in our opinion, not the raw time since menarche but the number of cycles experienced can really matter. Our data show that cervicovaginal aplasia carries a high risk of development of secondary endometriosis, but an early correction without restenosis favors a resolution of the lesions (we observed a relief of symptoms without recurrence in 4/5 (80%) adolescents).

There is limited evidence of endometriosis in patients with the unicornuate uterus [20]. In a study by Piriyev and Römer [21], histologically confirmed endometriosis (peritoneal, deep infiltrating, or ovarian) was detected in 87.5% of adult patients with a unicornuate uterus with an obstructed rudimentary horn with endometrium. It was significantly less prevalent in cases without active endometrium in the rudimentary horn (25%). In our group of adolescents with a unicornuate uterus with a non-communicating functional horn, the percentage of subjects who had endometriosis was also high (6/8, 75%). Following the treatment, complete resolution of dysmenorrhea was achieved in 37.5% of our patients with this malformation, whereas the other patients required long-term hormonal treatment of dysmenorrhea. However, the persistence of pain does not have to mean a recurrence of endometriosis. In a study published by Roman [22], adolescents, that underwent repeat laparoscopy because of persistent pelvic pain two years after surgical removal of endometriosis were found to have no visual or histological evidence of endometriosis.

In a systemic review by Jansen at al. [23], the overall prevalence of endometriosis, diagnosed by laparoscopy in adolescents with chronic pelvic pain, was 62%. In girls resistant to treatment with NSAIDS or oral contraceptive pills, the prevalence appeared to be 75% [23]. In our case series, the prevalence of endometriosis in patients with dysmenorrhea requiring medical assistance after surgical treatment of obstructive genital malformation reached 100%, but the number of patients was low. Further studies are necessary to check whether adolescents with dysmenorrhea that is resistant to NSAIDS and with a history of repair of an obstructive anomaly of the genital tract are at higher risk of developing endometriosis than a general population of adolescents with the same symptoms. There are several limitations to our study. Though the total number of patients was high, despite the low prevalence of obstructive Müllerian anomalies in the general population of women or even in patients with Müllerian anomalies [3], the number of patients treated for the same anomaly of the genital tract (in most cases there were complex malformations) was low, which limited the ability to conduct comparative statistical analysis between the subgroups. The second limitation concerns our diagnostic procedures to confirm or rule out endometriosis. As already mentioned in the methodology, a positive or negative observation for endometriosis was not always based on diagnostic laparoscopy with histology, which could have affected the results. The third limitation is the absence of a control group of age-matched adolescents without obstructive genital tract anomalies. Therefore, we could only refer to the limited published data on the prevalence of endometriosis in the general population of adolescents [14,23]. Finally, at least in some patients, the observation time was relatively short. In six patients with no evidence of endometriosis at the time of corrective surgery, but who complained of persistent dysmenorrhea resistant to NSAIDS, we diagnosed endometriosis (including moderate to severe disease in half of them) after 2.1–6.1 (median 3.8) years. We must acknowledge that more extended follow-ups could change these results.

Further studies are necessary to answer the questions that arise, for example: What are the determinants of the course of endometriosis in adolescents treated for obstructive Müllerian anomalies post-menarche? Do adolescents who are diagnosed with endometriosis at the surgical correction of obstructive Müllerian anomalies require long-term hormonal treatment to prevent the progress of the disease? Is it reasonable to perform laparoscopy simultaneously with vaginal septum resection in all post-menarcheal adolescents with OHVIRA syndrome?

## 5. Conclusions

Endometriosis occurs in about half of adolescents that undergo surgical treatment of obstructive Müllerian anomalies after menarche. Advanced endometriosis is present in nearly one-third of the affected girls. The incidence of endometriosis is highest in girls with cervical aplasia, in contrast with adolescents with an isolated transverse vaginal septum, who seem not to be at risk. The risk of developing endometriosis decreases after the surgical correction of obstruction in adolescence. However, it is still significant in girls with uterine anomalies, who are also at risk of persistent dysmenorrhea.

## Figures and Tables

**Table 1 jcm-12-02007-t001:** Characteristics of 15 patients diagnosed with congenital obstructive anomalies of the genital tract associated with cryptomenorrhea (menstruation without external bleeding). The anomalies are categorized according to the European Society of Human Reproduction and Embryology and the European Society for Gynecological Endoscopy (ESHRE/ESGE) classification system of female genital tract congenital anomalies [10], and the American Society for Reproductive Medicine Müllerian anomalies classification (ASRM MAC) 2021 [11]. Endometriosis is staged according to the revised ASRM (rASRM) classification of endometriosis [12]. CHC: combined hormonal contraception, aGnRH: GnRH agonist, US: ultrasound, MRI: magnetic resonance imaging, R: right, L: left, ASD: atrial septal defect, VSD: ventricular septal defect, and ?: unknown.

No.	Date of Surgery	Age at Surgery (Years)	Age at First Evaluation/ Symptoms Onset (Years)	Coexisting Anomalies	Treatment (Surgery)	Endometriosis (In Brackets rASRM Stage)
ESHRE/ESGE: normal uterus and cervical and vaginal aplasia (U0C4V4); ASRM MAC 2021: cervical agenesis.						
1	November 2016	13.6	11	-	Preoperative treatment with CHC/aGnRH. Creation of neovaginal canal, fenestration of the cervix, and cervicovestibular anastomosis (laparoscopy).	Yes (1). Peritoneal.
2	July 2017	15.3	14.6	-	Creation of neovaginal canal, fenestration of uterine body, and utero-vestibular anastomosis, (laparoscopy and laparotomy).	No at first surgery. Yes US/MRI 01.2020 (3). Endometrioma Ø40 mm left ovary.
3	January 2019	14.9	13	R kidney hypoplasia, ASD II, and VSD.	Creation of neovaginal canal, uterine body fenestration, and utero-vestibular anastomosis (laparoscopy and laparotomy).	Yes (3). Peritoneal, adhesions, and endometrioma Ø40 mm left ovary. Cystectomy.
4	September 2019	14.6	?	-	Creation of neovaginal canal, uterine body fenestration, and utero-vestibular anastomosis (laparoscopy and laparotomy).	Yes (2). Peritoneal and ovarian. Endometriosis coagulation.
5	December 2020	17.4	15.4	Partial septate uterus.	Creation of neovaginal canal, uterine cervix fenestration, and cervicovestibular anastomosis (laparoscopy).	Yes (2). Adhesions and culdesac obliteration. Adhesiolysis.
ESHRE/ESGE: bicorporeal uterus, normal cervix, and vaginal aplasia (U3C0V4); ASRM MAC 2021: uterus didelphys and distal vaginal aplasia.						
6	August 2016	16	15.3	Polycystic kidney disease.	Creation of neovaginal canal, fenestration of proximal vagina, and vagino-vestibular anastomosis.	No at first surgery. Yes laparoscopy September 2018 (4). Peritoneal, ovarian, culdesac obliteration, and adhesions. Adhesiolysis, endometriosis excision, and coagulation.
ESHRE/ESGE: normal uterus, normal cervix, and vaginal aplasia (U0C0V4); ASRM MAC 2021: distal vaginal aplasia.						
7	July 2018	13.2	0.1 *	Renal hypoplasia and cleft palate.	Creation of neovaginal canal, fenestration of proximal vagina, and vagino-vestibular anastomosis.	No.
8	March 2021	12.2	11.1	-	Creation of neovaginal canal, fenestration of proximal vagina, vagino-vestibular anastomosis, and laparoscopy.	Yes (2). Peritoneal and adhesions. Endometriosis coagulation and adhesiolysis.
ESHRE/ESGE: normal uterus, normal cervix, and transvers vaginal septum (U0C0V3); ASRM MAC 2021: transverse vaginal septum.						
9	May 2016	11.9	6.7 menarche 11.2016	L vesico-ureteral reflux.	Septectomy.	No.
10	June 2016	14.3	14.2	-	Septectomy.	No.
11	August 2018	16.3	?	-	Septectomy.	No.
12	November 2019	12.9	?	-	Septectomy.	No.
13	August 2020	13.1	11	-	Septectomy.	No.
14	October 2020	14.3	14.2	-	Septectomy and laparoscopy.	No.
15	June 2021	12.25	12.25	-	Septectomy.	No.

* In the first surgery of patient N 9 (hymenectomy and mucocolpos evacuation in laparotomy) was performed at the age of 1 month.

**Table 2 jcm-12-02007-t002:** Characteristic of 35 patients diagnosed with congenital obstructive anomalies of the genital tract in the presence of menstrual cycles. The anomalies are categorized according to the European Society of Human Reproduction and Embryology and the European Society for Gynecological Endoscopy (ESHRE/ESGE) classification system of female genital tract congenital anomalies [10], and the American Society for Reproductive Medicine Müllerian anomalies classification (ASRM MAC) 2021 [11]. Endometriosis is staged according to the revised ASRM (rASRM) classification of endometriosis [12]. OHVIRAS: obstructed hemivagina ipsilateral renal anomaly syndrome, R: right, L: left, ACUM: accessory cavitated uterine mass, and ?: unknown.

No.	Date of Surgery	Age at Surgery (Years)	Age at Menarche (Years)	Coexisting Anomalies	Treatment (Surgery)	Endometriosis (In Brackets rASRM Stage)
OHVIRAS ESHRE/ESGE: bicorporeal uterus, double cervix, and longitudinal obstructing vaginal septum (U3C2V2); ASRM MAC 2021: uterus didelphys and obstructed hemivagina.						
16	March 2012	11.75	11.2	R renal agenesis.	Laparotomy January 2012 Septectomy.	No at first surgery. Yes, second laparoscopy April 2018 (2). Peritoneal and adhesions. Adhesiolysis and endometriosis coagulation.
17	October 2012	12.25	11.5	R renal agenesis and scoliosis.	Septectomy.	Menstrual right tight pain. No MRI. No second laparoscopy August 2017.
18	January 2013	17.75	15	R renal agenesis.	Septectomy *.	No.
19	October 2013	16.7	11.5	L renal agenesis.	Septectomy *.	No.
20	May 2014	13	12.5	R renal agenesis.	Septectomy *.	No.
21	December 2014	13.1	11.4	L renal agenesis.	Septectomy *.	No.
22	December 2014	12.9	12.5	R renal agenesis.	Laparotomy December 2014 and septectomy.	Yes (3). Peritoneal and adhesions.
23	October 2015	12.7	11.9	R renal agenesis.	Laparoscopy July 2015 and septectomy.	No at first surgery. Yes, second laparoscopy February 2020 (2). Peritoneal and adhesions. Adhesiolysis and endometriosis coagulation.
24	December 2015	12.9	11	L renal agenesis.	Septectomy.	No at first surgery. Yes, second laparoscopy July 2019 (1). Peritoneal. Endometriosis coagulations.
25	April 2016	14.1	13.2	L renal agenesis.	Septectomy.	No.
26	June 2016	13.1	12.6	R renal agenesis.	Septectomy.	No.
27	November 2016	13.3	12.3	R renal agenesis.	Septectomy.	No.
28	July 2017	12.5	?	L renal agenesis.	Septectomy and laparoscopy.	Yes (2). Peritoneal and adhesions.
29	December 2017	12.2	11.8	R renal agenesis.	Septectomy and laparoscopy.	Yes (2). Peritoneal.
30	February 2018	16.3	15.8	R renal agenesis and vesico-urethral reflux.	Septectomy and laparoscopy.	Yes (3). Peritoneal, adhesions, and endometrioma Ø50 mm left ovary. Adhesiolysis and cystectomy.
31	February 2018	17.1	?	R renal agenesis.	Septectomy and laparoscopy.	No.
32	April 2018	12.3	12.3	L renal agenesis.	Septectomy.	No.
33	May 2018	11.4	?	L renal agenesis.	Septectomy and laparoscopy.	Yes (1). Peritoneal.
34	May 2019	12.6	12.2	L renal agenesis myelomenin-gocele and hydrocephalus.	Septectomy.	No.
35	June 2019	13.75	12.25	L renal agenesis and vesico-urethral reflux.	Septectomy and laparoscopy.	Yes (4). Adhesions, culdesac obliteration, and endometrioma Ø50 mm left ovary. Adhesiolysis and cystectomy.
36	August 2020	13.3	12.2	R renal agenesis.	Septectomy and laparoscopy.	No.
37	December 2020	14.3	14.	L renal agenesis.	Septectomy and laparoscopy.	Yes (3). Peritoneal, adhesions, and superficial sigmoid colon. Adhesiolysis and endometriosis excision.
38	May 2021	12.8	?	L nephrectomy.	Septectomy and laparoscopy.	No.
ESHRE/ESGE: bicorporeal uterus, double cervix, and longitudinal obstructing vaginal septum (U3C2V2); ASRM MAC 2021: uterus didelphys and obstructed R hemivagina.						
39	June 2020	12.4	12.2	-	Septectomy and laparoscopy (excision of left paraovarian serous cyst).	No.
ESHRE/ESGE: complete septate uterus, septate cervix, and longitudinal obstructing vaginal septum (U2C1V2); ASRM MAC 2021: obstructed R hemivagina and complete septate uterus with duplicated cervices.						
40	October 2017	13.3	12.0	-	Septectomy and laparoscopy.	No.
ESHRE/ESGE: bicorporeal uterus, unilateral cervical aplasia, and normal vagina (U3C3V0); ASRM MAC 2021: uterus didelphys with unilateral cervical atresia.						
41	May 2019	18.5	15.4 #	R renal agenesis and L vesico-ureteral reflux.	Right uterus excision and ipsilateral salpingectomy (laparoscopy and laparotomy).	No.
ESHRE/ESGE: hemiuterus with rudimentary cavity in not communicating horn (U4aC0V0); ASRM MAC 2021: unicornuate uterus with uterine remnant with functional endometrium.						
42	June 2012	18.2	12.2	R renal agenesis.	Rudimentary horn excision and ipsilateral salpingectomy (laparoscopy).	Yes (1). Peritoneal.
43	July 2015	17.6	11.8	-	Rudimentary horn excision and ipsilateral salpingectomy (laparoscopy).	Yes (1). Peritoneal. Endometriosis coagulation.
44	April 2016	13	11.9	R renal agenesis.	Rudimentary horn excision and ipsilateral salpingectomy (minilaparotomy).	No at first surgery. Yes, second laparoscopy September 2018 (3). Endometrioma Ø40 mm right ovary and peritoneal. Cystectomy and endometriosis coagulation.
45	April 2017	14.7	12.4	-	Rudimentary horn excision and ipsilateral salpingectomy (laparoscopy).	Yes (1). Adhesions. Adhesiolysis.
46	April 2017	18.1	10.5	Longitudinal non-obstructing vaginal septum and VACTERL association.	Rudimentary horn excision and ipsilateral salpingectomy (laparoscopy).	No.
47	April 2019	14.75	11.0	L renal agenesis.	Rudimentary horn excision and ipsilateral salpingectomy (laparoscopy).	Yes (1). Peritoneal.
48	October 2019	15.4	14.8	-	Rudimentary horn excision and ipsilateral salpingectomy (laparoscopy).	No.
49	August 2020	17.3	12.3	L renal agenesis.	Rudimentary horn excision (laparoscopy).	Yes (1). Peritoneal.
ACUM ESHRE/ESGE: unclassified anomaly; ASRM MAC 2021: variant.						
50	October 2017	14.6	13.0	-	Accessory cavitated uterine mass excision (laparoscopy).	No.

* In patients N 18–21 spontaneous microperforation of vaginal septum took place prior to the surgery. # Patient no. 41 had secondary hypothalamic amenorrhea for 3 years from menarche up to the surgery.

**Table 3 jcm-12-02007-t003:** Characteristics of 17 adolescents diagnosed with endometriosis at the time of surgical correction. N: number; M(−)—in adolescents with cryptomenorrhea; and M(+)—in adolescents with menses.

N of Patients/Total N of Patients with Anomaly	Age at Surgery (Years)	Time Symptoms Onset M(−)/Menarche M(+)–Surgical Correction (Years)	N of Patients with Minimal or Mild Endometriosis (Stage 1 or 2)	N of Patients with Moderate or Severe Endometriosis (Stage 3 or 4)	N of Patients with Persistent Dysmenorrhea
ESHRE/ESGE: normal uterus, cervical and vaginal aplasia (U0C4V4); ASRM MAC 2021: cervical agenesis.					
4/5 (80%)	13.6–17.4	1.9–2.6	3	1	0
ESHRE/ESGE: normal uterus, normal cervix, and vaginal aplasia (U0C0V4); ASRM MAC 2021: distal vaginal aplasia.					
1/2 (50%)	12.2	1.1	1	0	0
OHVIRAS ESHRE/ESGE: bicorporeal uterus, double cervix, and longitudinal obstructing vaginal septum (U3C2V2); ASRM MAC 2021: uterus didelphys and obstructed hemivagina.					
7/23 (30.4%)	11.4–16.3	0.4–1.25	3	4	4
ESHRE/ESGE: hemiuterus with rudimentary cavity in not communicating horn (U4aC0V0); ASRM MAC 2021: unicornuate uterus with uterine remnant with functional endometrium.					
5/8 (62.5%)	14.7–18.2	2.3–6	5	0	4

**Table 4 jcm-12-02007-t004:** Characteristics of six adolescents diagnosed with endometriosis during follow-up. N: number; M(−)—in adolescents with cryptomenorrhea; and M(+)—in adolescents with menses.

N of Patients/N of Patients with Anomaly and No Endometriosis at Correction	Age at Surgery (Years)	Time Symptoms Onset M(−)/Menarche M(+)–Surgical Correction (Years)	Age at Diagnosis of Endometriosis (Years)	N of Patients with Minimal or Mild Endometriosis (Stage 1 or 2)	N of Patients with Moderate or Severe Endometriosis (Stage 3 or 4)
ESHRE/ESGE: normal uterus, cervical and vaginal aplasia (U0C4V4); ASRM MAC 2021: cervical agenesis.					
1/1 (100%)	15.3	0.7	17.75	0	1
ESHRE/ESGE: bicorporeal uterus, normal cervix, and vaginal aplasia (U3C0V4), ASRM; MAC 2021: uterus didelphys and distal vaginal aplasia.					
1/1 (100%)	16	0.7	18.1	0	1
OHVIRAS ESHRE/ESGE: bicorporeal uterus, double cervix, and longitudinal obstructing vaginal septum (U3C2V2); ASRM MAC 2021: uterus didelphys and obstructed hemivagina.					
3/16 (18.75%)	11.75–12.9	0.6–1.9	16.9–17.8	3	0
ESHRE/ESGE: hemiuterus with rudimentary cavity in not communicating horn (U4aC0V0); ASRM MAC 2021: unicornuate uterus with uterine remnant with functional endometrium.					
1/3 (33.3%)	13.0	1.1	15.4	0	1

**Table 5 jcm-12-02007-t005:** Comparison of subgroups of the subjects. N: number; M(−)—in adolescents with cryptomenorrhea; and M(+)—in adolescents with menses.

	Girls with (N = 23) vs. without Endometriosis (N = 27)		Girls Diagnosed with Endometriosis at Surgery (N = 17) vs. during Follow-Up (N = 6)		Girls with (N = 18) vs. without Persistent Dysmenorrhea (N = 32)	
	Mean	*p* Value	Mean	*p* Value	Mean	*p* Value
Age at surgery (years)	14.4 vs. 14.2	0.872	14.6 vs. 13.6	0.400	14.6 vs. 14.1	0.530
Age at symptoms onset M(−)/menarche M(+) (years)	12.6 vs. 12.7	0.626	12.6 vs. 12.7	0.772	12.6 vs. 12.6	0.653
Delay symptoms onset M(−)/menarche M(+)–surgery (years)	2.0 vs. 1.5	0.249	2.4 vs. 1.0	0.231	2.2 vs. 1.6	0.377
Aga at last evaluation (years)	17.2 vs. 16.1	0.208	16.8 vs. 18.4	0.068	18.7 vs. 15.8	0.004
Time surgery-last evaluation (years)	3.4 vs. 1.9	0.027	2.9 vs. 4.8	0.099	4.4 vs. 1.9	0.001

## Data Availability

Not applicable.

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
