# Peer review of "Endometriosis in Adolescents with Obstructive Anomalies of the Reproductive Tract"

_jcm, 2023, doi:10.3390/jcm12052007_

Round 1

Reviewer 1 Report

Endometriosis in adolescent is emerging as a key topic in the pathogenesis and treatment of this disease. The present paper, however, does not add any new knowledge, since it is known that patients with obstructive anomalies have higher risk of endometriosis (and indeed explained in the discussion about previously published papers with much bigger case series). 

The main criticism is the absence of a control group of healthy patients of the same age, which makes impossible to run any statistical analysis about prevalence of endometriosis or its associated symptoms. 

Table 1 (the only provided table) presents row data and it is impossible to extrapolate any information from it. The authors should provide summary tables in which mean values of each parameter are presented, in order to describe their population and allow reader to draw conclusions. Moreover, a table of the clinicals variables considered (pelvic pain, need for painkiller...) should be also added. 

It would be interesting to run analyses between pain symptoms before surgery (which can be explained by the obstruction itself) and after surgery (related to endometriosis). Another idea would be to correlate the localization of the pelvic endometriotic lesions (or ovarian lesions) with the type of obstruction.

None of this was done. The authors should work on their data to evidence new information about endometriosis and obstructive anomalies in adolescents. This may be hampered by the low number of patients falling into each subcategory of obstructive anomaly (as they are all quite rare).

Author Response

Dear Reviewer,

Thank You very much for your time and for the constructive and valuable revision of our manuscript.  

According to your suggestions we have done many changes to the manuscript. We have added two tables and we have extended the analysis and the discussion of our results. 

Remark 1. The previously published papers, presented in the discussion, involve case series with larger groups of patients, but the groups of subjects involve adult women, not exclusively adolescents as in our study.

Remark 2. We are aware that the  comparison with the control group of age-matched adolescents without obstructive genital tract anomalies would have great value but some barriers must be considered. It would be possible to evaluate 50 age matched adolescent with ultrasound or MRI but from ethical point of view it would be rather unacceptable to perform laparoscopy (that is usually necessary to diagnose or rule out endometriosis in this age group) without medical indications.  It may also explain why the data on endometriosis in general population of adolescents are lacking. We have referred to the papers published by Rasp E (2022) and Song XC (2020).

Remark 3. The data presented in the Tables 1 and 2 have been summarized in the text (section Results).  In response to your suggestions we have added two tables. 

Reviewer 2 Report

1. Table 1 and table 2 should be presented in the result section rather than method section. 

2. abbreviations in tables need to clarified in each table 

Author Response

Dear Reviewer,

Thank You very much for your time and for the revision of our manuscript. 

We have corrected the tables, explained all the abbreviations used and removed the tables from the materials and methods section to the results section.

Reviewer 3 Report

This is a collation of cases with uterovaginal anomalies causing obstruction of menstrual flow that are presented with reference to the occurrence of endometriosis. The authors identified and acknowledged the methodological limitations. Many of these are related to the difficulties of studying rare and infrequent diseases and because of the need for invasive testing to diagnose endometriosis. Despite the limitations the study presents an interesting case series and, as such, is worthy of publication.

The article would benefit from revision to address the following points.

line 31: appear: you mean manifest/ or are diagnosed.

Table 1: use of decimal points in relation to age: please use (.) instead of (,) also, it is unclear what the decimal point refers to: is this a fraction of the year? what is for example 12,25? 

Patient 15 had surgery at the same age as appearance of symptoms, please verify.

0,1 is referred to in the text as 1 month?

The years appear in one or two decimal points: perhaps clearer if the data

was presented as year and month.

Explain CHC if legend

line 72: you means from menarche?

Is creation of neovaginal canal, fenestration of proximal vagina the same as vagino-vestibular anastomosis? 

line 92: 0.1 year follow-up? is that meaningful?

line 194: but developed three years later. rephrase: developed what?

line 197: in this case...etc. please correct grammar.

line 207: the proportion...etc. please correct grammar.

line 211. in the study of Roman ...etc. please correct sentance grammer.

line 218: possibility: you mean ability?

line 226: supervision: you mean follow-up?

line 238: contrary: you mean: in contrast?

Can you provide an analysis of the occurrence of endometriosis in relation to the time interval from menarche to diagnosis?                                                   

Author Response

Dear Reviewer,

Thank You very much for your time and for the constructive revision of our manuscript.

We have done all the changes to the manuscript according to your suggestions.  We performed the English revision.

The age/time from symptom onset or menarche to surgery was calculated with an accuracy of one month and presented in years (0.1 is one moth, 0.25 is equal to 3 months). We corrected the tables.

Round 2

Reviewer 1 Report

I agree with the authors that it is not always possible to have a control group, like in this case and that the series is interesting as it involves a very young group of adolescents and aims to address an important issue in our understanding of endometriosis. Presentation of data and discussion is still lacking of scientific soundness, as the series it is presented as a long list of different cases, and the lack of any analysis hampers the possibility to draw conclusions and have appropriate comparison with the existing literature. After having read authors answers and the edited version of the manuscript, I still have major comments:  

-          Regarding comparison with healthy controls: Please include the justification for absence of control group in adolescents in the discussion (i suggest at the beginning, after stating that there is not a lot of data on the exact prevalence of endometriosis in adolescents).

-          Tables 2 and 3 help to better understand data, but still there is no statistical analysis at all. A x-square (o Fisher test) could be done to compare the presence of endometriosis in patients suffering from pelvic pain before diagnosis and also the persistence of pain after surgery in patients diagnosed or not with endometriosis (in this case there is no big limitation due to individual very small subgroups).

-          All statements ‘much lower incidence in OHVIRAS’ (line 192), ‘significantly reduced risk of developing advanced endometriosis’ (line 197), ‘much higher occurrence of endometriosis of 100%’ (line 222) ‘significantly lower but also high percentage of subjects who had endometriosis’ (line 246), and so on are all overstatements, as there is no real statistical analysis but only descriptive results. The authors should change the way they discuss their data and compare them to the literature, unless they add statistical analyses that confirm these data.

-          Line 238: talking about risk of recurrence should be done with caution, as the follow up of this series is not long, and patients may show recurrence later over the years.

-          Lines 246-251: pain recurrence/persistence after surgery and presence of endometriosis is really an interesting pint emerging from data, but the absence of analyses does not allow any conclusion.

Author Response

Dear Reviewer,

We are very grateful for Your time and effort to revise our manuscript. We also appreciate Your patience.

Remark 1: We have added  the information about the absence of the control group of age-matched adolescents to the limitations of our study.

Remark 2: We have performed statistical analysis and compared adolescents with persistent dysmenorrhea to adolescents without persistent dysmenorrhea, the subjects with endometriosis to the subjects without endometriosis  as well as the subjects  with endometriosis diagnosed at surgery to the subject with endometriosis diagnosed in the period of follow-up. We do realize that the last should be interpreted with caution as the subgroups were small (17 vs 6). We added Table 5.

Remark 3: We have done all the necessary changes to the discussion according to the suggestions.

Remark 4: We are aware that extended follow-up could change the results . We have discussed it in the limitations of our study. 

English revision has been made.